# Animal Trypanosomiasis: Challenges and Prospects for New Vaccination Strategies

**DOI:** 10.3390/microorganisms12122575

**Published:** 2024-12-13

**Authors:** Samille Henriques Pereira, Felipe Paladino Alves, Santuza Maria Ribeiro Teixeira

**Affiliations:** 1Departamento de Bioquímica e Imunologia, Universidade Federal de Minas Gerais, Belo Horizonte 31270-901, MG, Brazil; samillehenriques@gmail.com (S.H.P.); felipepaladino@ufmg.br (F.P.A.); 2Centro de Tecnologia em Vacinas, Universidade Federal de Minas Gerais, Belo Horizonte 31310-260, MG, Brazil

**Keywords:** animal trypanosomiasis, nagana, surra, dourine, vaccine, mRNA vaccine, vector-based vaccine, CRISPR-attenuated parasite vaccine

## Abstract

Animal trypanosomiasis, such as nagana, surra, and dourine, represent a significant challenge to animal health and economic development, especially in tropical and subtropical regions where livestock production is an essential component of a country’s economy. Despite advances in the control of human trypanosomiasis, animal diseases caused by several species of trypanosomes remain neglected. The lack of funding for the development of new treatments and vaccines contributes to sustaining the severe economic impacts these diseases have on the farming industry, especially in low-income rural areas. Recent advances in the understanding of the immune processes involved during infection have been essential for the development of new approaches towards disease control including vaccines. These new approaches must be part of integrated control programs, which must also include vector management and the awareness of good veterinary practices. Addressing the challenges posed by the control of animal trypanosomiasis requires collaborative and continuous efforts shared among scientists, governments, and the farming industry, if significant progress is to be made to mitigate the impact of these diseases. In this literature review, we discuss the main challenges for the development of vaccines for animal trypanosomiasis and the research underway, including the prospects for employing new vaccine platforms, such as an mRNA vaccine, vector-based vaccine, and CRISPR-attenuated parasite vaccine.

## 1. Introduction

Animal trypanosomiasis refers to diseases known as nagana, surra and dourine, caused by single-celled protozoan parasites of the genus *Trypanosoma*. In many countries, these diseases have a significant impact on beef and dairy farming [1]. The parasites are transmitted mainly by insect vectors and affect different animals, including cattle, goats, pigs, and horses [2]. Trypanosomes have caused diseases in farm animals throughout history, with records dating back centuries. In the first records dating to the 19th century, veterinarians observed disease-debilitated herds in Africa and other parts of the world. In 1895, David Bruce established the relationship between nagana trypanosomiasis (in cattle) in Africa and *Trypanosoma brucei* (*T. brucei*). At the same time, the tsetse fly was identified as responsible for transmission, which was fundamental for understanding the epidemiology of trypanosomiasis [3,4].

Parasites that cause animal trypanosomiasis have a life cycle that alternates between an invertebrate vector (tsetse flies or other hematophagous insects) and a vertebrate host (domestic or wild animals as well as humans) (Figure 1). The cycle begins when the vector bites an infected animal and ingests bloodstream trypomastigotes from the host. In the vector’s intestine, the trypomastigote forms differentiate into procyclic forms that multiply by binary division in the insect’s digestive tract. The procyclic forms then migrate to the salivary glands (in the case of *T. brucei* and *Trypanosoma congolense* (*T. congolense*)) or to the proboscis (*Trypanosoma vivax* (*T. vivax*)), where they transform into metacyclic trypomastigotes, which are the infective forms of the parasite. When the infected vector bites another animal, the metacyclic trypomastigotes are inoculated into the bloodstream of the new host. In the host bloodstream, the metacyclic trypomastigotes quickly transform into blood trypomastigotes, the form that multiplies and circulates through the blood and other tissues, such as lymph nodes and cerebrospinal fluid. After multiplying by binary division and invading different tissues, trypomastigotes cause different symptoms of trypanosomiasis. Because some trypomastigotes remain in the bloodstream, they are able to infect another vector when the animal is bitten again [5,6].

In many regions, the control of animal trypanosomiasis has been highly challenging. In contrast, the control of human trypanosomiasis, or sleeping sickness, caused by *T. brucei*, has made significant progress in recent years mainly due to the development of new drugs and the implementation of control programs in endemic areas. Animal trypanosomiasis continues to be a highly neglected issue mainly because of limited resources and global attention directed towards disease control. As with other diseases that affect humans, sleeping sickness attracted better research funding compared to animal trypanosomiasis. The economic incentives for developing veterinary drugs for cattle disease are often low, resulting in limited options for treatment. Although animal trypanosomiasis has a significant economic impact, especially in rural and developing regions, affecting the production of meat, milk, and other products, the scale of this impact tends to be underestimated compared to the direct effects on human health [7,8,9,10,11,12].

Since the 1970s, it is estimated that around 35 million doses of trypanocides are used every year in many regions of the world. The main treatments depend on diminazene aceturate, which has a curative action; homidium bromide and chloride, which are curative and have also some prophylactic actions; and isometamidium, which is curative and has also a strong prophylactic action. Despite the high number of doses applied, this number covers only one-third of the cattle at risk [6]. In addition, the emergence of drug resistance has complicated the management of these diseases, making it necessary to develop new therapeutic and control strategies [8,13]. Although advances in the development of new drugs for animal trypanosomiasis are still limited, some new promising compounds, such as benzimidazole derivatives, have been proposed [14]. Acoziborole, originally developed for human trypanosomiasis, has shown the potential to treat animal infection cause by several *Trypanosoma* species, but its veterinary application is still in the experimental stages [15]. Other drugs initially developed for human trypanosomiasis, which have been tested in pre-clinical studies as a possible treatment for trypanosomiasis in cattle, include fexinidazole [16] and pafuramidine [17]. In addition to the development of new experimental drugs, much effort has been focused on modifying or combining existing drugs, such as diminazene and isometamidium, to increase their effectiveness and combat resistance [18].

In addition to effective treatment, the correct diagnosis of diseases caused by trypanosomes is highly important, particularly in regions where several species of trypanosomes coexist. The accurate differentiation among species is essential for the proper treatment and the implementation of adequate control measures [19]. A clinical diagnosis is based on symptoms presented by the animals, such as anemia, weight loss, fever, and lymphadenopathy. However, these are non-specific signs that can be confused with other diseases, and it is impossible to differentiate among trypanosomes. The most widely used method of diagnosis is microscopy, in which Giemsa staining is used to visualize trypanosomes in blood smears. When the micro-hematocrit method is used, blood is centrifuged to help the visualization of parasites under the microscope [20,21]. These methods have low sensitivity, especially with low parasitemia infections, making early detection difficult. Among the serological methods, indirect immunofluorescence (IFI) and enzyme-linked immunosorbent assay (ELISA), which detect specific antibodies in the serum from infected animals, are highly useful for diagnosis including in cases of subclinical trypanosomiasis [22,23]. Although these techniques are more easily executed compared to molecular methods, they can lead to false-positive results due to cross-reaction with antigens from other parasites. Molecular methods, based on polymerase chain reaction (PCR), are highly specific for the identification of trypanosome subspecies. Although loop-mediated isothermal amplification (LAMP) offers the possibility to be used as a point-of-care molecular test, it is still not widely used [23,24]. The development of new molecular and serological tests with increased sensitivity and specificity for trypanosomes of interest to livestock using DNA sequences or recombinant antigens is highly warranted. The implementation of more robust surveillance programs depends on the use of a combination of different diagnostic methods, and this important issue has been thoroughly addressed in recent reviews [25,26].

Together with new efforts towards drug development and cost-effective diagnostic tests, the development of preventive or therapeutic vaccines for animal trypanosomiasis is an essential step towards the control of these diseases. However, as for all diseases caused by protozoan parasites, vaccine development faces major challenges, including the design of the immunogens, which is a highly complex issue due to the ability of trypanosomes to evade the host immune system. Here, we reviewed the literature with the aim of providing a summary of the major impacts caused by these diseases and the current status of the main control methods as well as discussing the key challenges and difficulties faced by researchers involved with the development of vaccines for animal trypanosomiasis. We also present new and exciting perspectives for the use of next-generation vaccine platforms that could help overcome these barriers. To achieve this, we (i) searched the PubMed, ScienceDirect, and Google Scholar databases for articles published up to October 2024 using the keywords: animal trypanosomiasis, livestock trypanosomiasis, *Trypanosoma vivax* vaccine, *Trypanosoma brucei* vaccine, *Trypanosoma congolense* vaccine, *Trypanosoma equiperdum* vaccine, and *Trypanosoma evansi* vaccine, and (ii) selected relevant studies published in indexed journal articles and discussed their main findings. The selected sources were summarized and synthesized in this literature review.

## 2. Different Trypanosome Species That Cause Animal Trypanosomiasis

Often referred to as African trypanosomiasis, some Trypanosoma species also cause animal disease outside Africa. The three main types of trypanosomiasis that stand out in livestock contexts are nagana, surra, and dourine. These conditions are of great concern in tropical and subtropical regions, where the interaction between hosts and vectors facilitates the spread of the parasites, affecting animals of different species. Table 1 shows the hosts of these diseases.

### 2.1. Nagana

Nagana has *T. congolense*, *T. vivax*, and *T. brucei brucei* (*T. b. brucei*) as its etiologic agents. The disease affects several species of domestic and wild animals in sub-Saharan Africa [27].

*T. vivax*, originally from Africa, is now present in Latin America and the Caribbean. It mainly infects cattle but affects sheep, goats, and other ruminants. It is the most phylogenetically distinct. Pathogenicity can vary in cattle where acute and hemorrhagic or subclinical infections can occur [25,28]. Typically, *T. vivax* was described as being confined to the host’s vascular system; however, Melo-Junior et al. (2024) demonstrated the presence of the protozoan in the acute phase in the lymph nodes, brain, cerebellum, lung, kidney, and liver. In the chronic phase, in addition to those found in the acute phase, they also observed *T. vivax* in the bone marrow, eyes, ear, uterus, adipose tissue near the testicle, epididymis, and testicle [23]. It is transmitted in Africa by the tsetse fly. Still, in Latin America and the Caribbean, it is transmitted mechanically by tabanids and other insects since *T. vivax* remains confined to the proboscis of the insect, where it completes its short life cycle and iatrogenic transmission through contaminated syringes that are used on various animal [23,29]. Although not recommended, the reuse of syringes is common in Latin America. Studies have shown that up to 30% of animals can be infected by contaminated needles via the subcutaneous route, 50% through the intramuscular route, and 80% through the intravenous route. In some drugs, as vaccines against foot-and-mouth disease, *T. vivax* can survive for up to 20 h in the syringe. Mechanical transmission has allowed *T. vivax* to spread far beyond the limits of the African tsetse belt [30]. The trypanosome transmission cycle is described in Figure 2.

*T. congolense* is distributed in Africa and is considered to be the most pathogenic species that infects cattle. It can also infect goats, pigs, and other ruminants and is transmitted by the tsetse fly. In the vertebrate host, the pathogen is confined to the vascular system, attaching its flagella to erythrocytes and endothelial cells, and damage is caused at the site of adhesion [31,32].

*T. brucei* is found predominantly in sub-Saharan Africa, affecting mainly cattle but also other domestic and wild animals. The tsetse fly transmits it. It has three known subspecies: *T. b. brucei*, which causes nagana in cattle, and *Trypanosoma brucei gambiense* (*T. b. gambiense*) and *Trypanosoma brucei rhodesiense* (*T. b. rhodesiense*), which are the causative agents of human African trypanosomiasis (sleeping sickness). The human disease is characterized by two stages: the early stage, known as hemolymphatic infection, and the late stage of meningoencephalitis infection. Mainly found in the vascular system, the parasite is also observed in other tissues including the brain [33,34].

### 2.2. Surra

*Trypanosoma evansi* (*T. evansi*) causes the disease of surra in Africa, Latin America, the Middle East, and Asia. It infects horses, camels, cattle, and pigs. Like *T. vivax*, it has developed a mechanism of mechanical transmission by tabanids and other biting insects, which has allowed this pathogen to be transmitted to places other than Africa through infected imported animals. Iatrogenic transmission is also reported, in addition to sexual transmission, oral contamination, and by hematophagous bats. *T. evansi* has no biological cycle in the vector, unlike the trypanosomes that cause nagana. The symptoms of the outbreak can vary in intensity depending on the host species and geographical area [35,36].

### 2.3. Dourine

Unlike other species that are transmitted by vectors, *Trypanosoma equiperdum* (*T. equiperdum*), another trypanosome of interest to livestock, is sexually transmitted. It is distributed in Europe, Asia, and South America and infects horses, causing coitus disease characterized by genital edema, skin lesions, and, eventually, paralysis and death. The most common symptoms are swelling of the genitals, weakness, weight loss, urethral discharge, and neurological syndromes. Dourine is usually fatal in untreated horses, while in donkeys and mules, it can have subclinical symptoms [37,38].

## 3. Challenges in Vaccine Development for Animal Diseases Caused by Trypanosomes

Despite the large economic impact trypanosomes have caused throughout decades, there are still no licensed vaccines. This is mainly due to a combination of factors, including those that can hinder the selection and testing of vaccine candidates. Antigenic variation, immunosuppression of host responses, and the limitations regarding laboratory animal models for testing vaccines are three important factors discussed in this review.

### 3.1. Host Immune Response and Immunoregulatory Pathways During Trypanosoma Infections

The host immune response during Trypanosoma infections is complex and involves both innate and adaptive pathways. To be able to survive in its host, the parasite uses sophisticated mechanisms to manipulate the host’s immunoregulatory pathways and evade immune defenses [39,40].

Different from *Trypanosoma cruzi*, which causes a human disease known as Chagas disease, trypanosome species that cause infections in livestock animals reside in the bloodstreams of their mammalian host and are therefore direct targets of antibody-mediated destruction. For the destruction of *T. brucei* by the trypanolytic factor in human serum, specific elements in human serum, such as apolipoprotein L1 (ApoL1) and related haptoglobin (Hpr), are essential. Once internalized, these components bind to the parasite and ApoL1 causes an osmotic imbalance in the parasite’s cells, leading to lysis. This trypanolytic activity is exclusive to human serum and some primates, which prevents other mammals, such us bovines, from developing this natural response to *T. brucei*, making them susceptible to infection [41].

As part of innate immune pathways that initiate an acute inflammatory response, different host cells are activated by various trypanosomal factors. During the acute phase of infection, cytokines such as TNF-α, IFN-γ, and IL-12 are produced in response to the parasite. They activate immune cells to fight the infection, but trypanosomes can negatively regulate the production of these cytokines to prevent an effective inflammatory response. Trypanosomes stimulate the production of anti-inflammatory cytokines, such as IL-10 and TGF-β, which suppress the host’s immune response. IL-10, in particular, inhibits the production of pro-inflammatory cytokines and limits the activation of T cells and macrophages. TGF-β also plays a critical role in inhibiting immune cell activation and promotes an immunosuppressive environment that favors parasite persistence [42,43,44].

M1 macrophages are activated in pro-inflammatory environments, producing high levels of cytokines such as TNF-α and IL-12, as well as nitric oxide, which favors the destruction of intracellular parasites. However, trypanosomes tend to inhibit M1 polarization, preventing this cytotoxic response. Trypanosomes promote the activation of M2 macrophages, which are associated with tissue repair, immunosuppression, and immune tolerance. M2 macrophages produce IL-10 and TGF-β, which limit inflammation and favor parasite survival. M2 polarization creates a permissive environment for chronic infection, facilitating immune evasion [45,46].

Dendritic cells (DCs) are essential for the activation of T lymphocytes, but trypanosomes can negatively modulate their activation. This prevents the proper presentation of antigens and the production of cytokines such as IL-12, which are necessary to direct the response of T helper type 1 (Th1) cells, which are critical in the defense against parasitic infections. The deactivation of dendritic cells by trypanosomes reduces the immune system’s ability to initiate an effective adaptive immune response [45].

Regulatory T lymphocytes (Tregs) are crucial for maintaining immune homeostasis, suppressing excessive responses and preventing damage to host tissue. During Trypanosoma spp. infection, Tregs are often induced or expanded to control inflammation. Activation of the Tregs results in the production of IL-10 and TGF-β, which suppress the proliferation of effector T cells, such as CD4^+^ and CD8^+^ T lymphocytes, which could eliminate the parasite [42]. This reduces the effectiveness of the adaptive immune response, allowing trypanosomes to escape immune surveillance and establish chronic infections. In addition, Tregs can influence the polarization of macrophage responses, favoring an M2-type response (immunosuppressive) rather than an M1-type response (pro-inflammatory), which contributes to the chronicity of the infection [43,45].

These immunoregulatory pathways represent a major challenge for the development of effective vaccines against animal trypanosomiasis. The suppression of adaptive immune responses and the induction of immunosuppressive environments hinder the generation of protective immunological memory, an essential element for effective vaccines. Future vaccination strategies therefore need to consider ways of bypassing or modulating these immunoregulatory pathways to induce a more effective and long-lasting immune response. The host’s immune response to animal trypanosomiasis was extensively covered in other reviews [39,47].

### 3.2. Parasite Antigenic Variation

One of the biggest obstacles in vaccine development for infections caused by trypanosomes is the mechanism of antigenic variation present in several species including *T. brucei*, *T. congolense*, and *T. vivax*. These parasites present variant surface glycoproteins (VSGs) that cover the surface of blood-stage trypanosomes. The VSG proteins are structured into three main functional units: the signal peptide, which directs the protein to the cell surface and is removed during the translocation process; an extensive and highly variable N-terminal domain (NTD), which makes up the bulk of the molecule; and a smaller C-terminal domain (CTD) with a glycosylphosphatidylinositol (GPI) anchor, responsible for the attachment of the VSG to the membrane. Because they are GPI-anchored, during infection, large numbers of VSG molecules are released into the circulation [48,49,50].

The parasite genome has hundreds of genes encoding different VSGs, but only one VSG is expressed at a time in a single parasite cell. When the host’s immune system recognizes these VSG proteins and produces specific antibodies that bind to the surface of every parasite expressing this VSG on its surface, these parasites are eliminated. Because some parasites are able to change the VSG gene that is actively expressed, these few parasites avoid destruction by the immune attack and multiply. Therefore, during an infection, different VSGs can be found covering the parasite surface. VSG gene expression switch occurs through genome recombination or by activating the expression of a silent VSG gene from a large gene repertoire present in the parasite’s genome. VSG switch can occur spontaneously during the parasite cell division, resulting in mixed populations with different antigenic variants that are exposed to the host immune system. This continuous variation in surface antigens is a highly sophisticated immune evasion mechanism that prevents a single vaccine from generating long-term protective immunity against all variants unless a highly immunogenic, non-variable antigen is selected as antigen candidate [51,52].

### 3.3. Immunosuppression of Host and Parasite Persistence

Trypanosomes use several strategies to suppress the host immune response, reducing the effectiveness of vaccines. Infection can lead to an ineffective immune response, where the immune system does not respond adequately to the antigens introduced by the vaccine [39]. Trypanosome infections can also affect the function of T and B lymphocytes, which are crucial for the immune response. Antibody production can be reduced, and the ability of T cells to recognize and eliminate pathogens can be compromised [53,54,55]. The parasite can cause dysregulation in the immune system, leading to a chronic inflammatory environment or a state of immunosuppression. This environment can make it difficult to induce and maintain an effective immune response after vaccination. Immunosuppression can result in short-term immunity, where the protection conferred by the vaccine is not long-lasting. This may require the administration of more frequent vaccines or the combination of vaccines with other forms of treatment to ensure effective protection [39]. African trypanosomes also avoid the host’s immune response by exploiting complement activation. They accomplish this by eliminating large amounts of VSG in the circulation, which forms immune complexes with antibodies and prevents the parasite from being lysed by complement. This strategy reduces the deposition of the attack complex on the membrane and can lead to a state of hypocomplementemia, which is common in African trypanosomiasis. In addition, VSG masks the binding of complement proteins, blocking the activation of the alternative complement pathway and thus preventing trypanosome lysis [56,57].

Parasites that can persist in the host cause chronic infections, causing continuous suppression or modulation of the immune system. As a result, the host may not mount a sustained immune response to eliminate the parasite, even after vaccination. Chronic infections also complicate the interpretation of vaccine efficacy studies, as the parasite may be present at low levels or in hidden niches, making it difficult to assess vaccine success. Some parasites can hide in immune-privileged niches within the host’s body, such as the central nervous system, where the immune response is less active. Persistence in these niches means that even if a vaccine manages to induce an effective immune response, the parasite can continue to survive and replicate in these protected sites, evading complete elimination [58,59,60].

### 3.4. Limitation of Effective Laboratory Models

The limited choices of animal models that mimic natural infection in farm animals is another challenge that makes it difficult to evaluate vaccine candidates. In contrast to *T. brucei*, *T. vivax* is difficult to cultivate in the laboratory, which hinders the development of immunogens and restricts biological studies with this parasite. Axenic cultivation has only been described a few times and is difficult to reproduce in other laboratories [61,62]. Most of the in vivo studies using laboratory models have been described with the Y486 *T. vivax* strain and its derivatives isolated in Africa and able to infect mice. Also using the Y486 strain, D’Archivio et al. (2011) described an in vitro cultivation protocol that allowed the genetic manipulation of the parasite, incorporating the luciferase enzyme [62]. This model was used to evaluate a vaccine candidate in a murine model [63]. Despite its relevance as a pathogen of bovines, studies on the physiology and biochemistry of *T. vivax* are scarce. It requires highly specific conditions, and the culture media available are often not capable of sustaining its growth for long periods. The axenic culture of blood trypomastigotes forms of *T. congolense* was possible for a few strains, such as IL3000 and STIB910 [64]. Some work demonstrated infection in a murine model, but only for specific strains [65,66]. This is in sharp contrast with other species that cause African Animal Trypanosomiasis, such as *T. b. brucei*, which is the most widely studied since most strains are well adapted in axenic cultures in the laboratory and are able to infect murine models.

Similar limitations occur in studies with *T. evansi*. Recently, Kamyingkird et al. (2022) described the isolation of Thai strains in culture on special media and in mice for the first time [26]. Tanaka et al. (2021) described an animal model for the IVM-t2 isolate of *T. equiperdum* and managed to reproduce pathology similar to that of horses, making it a valuable model for studies and the testing of vaccine candidates for this pathogen. A limitation of the model is its unsuitability for studies of sexual transmission, which is crucial in dourine and cannot be studied in these models [67].

Although animal models have provided studies of the general biology of parasites, immune system evasion, and cellular response, mainly with the species *T. brucei* and a few strains of *T. vivax* and *T. congolense* [68], these models may not fully replicate the conditions of infection in natural hosts, such as cattle. Significant differences in pathogenesis and immune response between mice and larger hosts make the pre-clinical testing of vaccines difficult. Despite that, laboratory models are crucial due to ethical and regulatory constraints, especially relevant for large models such as cattle, horses, and camels.

Unpublished data from our group showed that, despite great effort, it was not possible to establish the infection in BALB/c mice, IFN-γ knockout mice, or in guinea pigs with a *T. vivax*. The animals were inoculated with fresh or cryopreserved blood from goats infected with a *T. vivax* isolate obtained from a 2008 outbreak in the state of Minas Gerais, in Brazil. Inoculation took place intraperitoneally with 1 × 10^5^ parasites for mice and 7 × 10^6^ for guinea pigs. Parasitemia was monitored by blood smears for 30 days. Axenic cultivation was also attempted using fresh blood from infected goats. The blood was submitted by low centrifugation to remove red blood cells, and the supernatant was added to different culture media, including an LIT (liver infusion tryptose) medium commonly used to cultivate *T. cruzi*, HMI-1 and SMD-79 medium used to cultivate *T. brucei* as well as DMEM (Dulbecco’s Modified Eagle Medium). In none of these conditions was it possible to detect parasites after several days of culture.

## 4. Advances in Research Towards the Development of Vaccines Against Animal Trypanosomes

Advances towards the development of vaccines based on four different vaccine strategies will be discussed in the next section: vaccines based on recombinant antigens, mRNA vaccines, vaccines based on viral vectors, and vaccines based on genetically attenuated parasites. Each vaccine strategy relies on slightly different mechanisms to activate the host immune response, which, in turn, may result in distinct efficacies. When infecting its host, a Trypanosoma parasite carries many of the pathogen-associated molecular patterns (PAMPs), which are recognized by the pattern recognition receptors (PRRs) of the innate immune system, such as Toll-like receptors (TLRs). This triggers an initial response from innate immune cells, such as macrophages and dendritic cells, that initiate the production of inflammatory cytokines [69]. Vaccines based on attenuated parasites have the advantage of carrying all the components that can be presented by macrophages and other antigen-presenting cells (APCs) to T lymphocytes in the lymph nodes, to activate both CD4^+^ (helper) and CD8^+^ (cytotoxic) T lymphocytes [70]. As an important characteristic of attenuated vaccines, because they mimic a natural infection, these vaccines generally generate a robust immune response. This includes the production of memory T and B cells, which remain in the body for long periods after vaccination. To generate attenuated trypanosomes, CRISPR could be used to remove essential genes for antigenic variation, reducing the parasite’s ability to evade the immune response and enabling effective vaccination [69].

Vaccines based on viral vectors and mRNA vaccines share a similar mode of action. After the mRNA vaccine injection, the lipid nanoparticles deliver the mRNA into the body’s cells, typically muscle cells at the injection site. Once inside the cells, the mRNA does not enter the nucleus or alter the cell DNA but remains in the cytoplasm, where it begins to be translated by the ribosomes. The cell’s ribosomes read the mRNA and start producing the encoded protein [71]. After the protein is produced, the mRNA is naturally degraded within the cells in a short period, eliminating concerns about permanent genetic alterations [71,72]. For vector-based vaccines, the vector enters the body and infects cells, similar to a natural virus infection, but without causing the disease, as it has been genetically modified to be harmless. The vector delivers the pathogen’s genetic material into the host cell nucleus, where this material is transcribed into mRNA [71]. The host cells use the pathogen’s DNA or RNA to produce a specific protein (the antigen) associated with the target pathogen. In both cases, the proteins generated are recognized by the immune system. The proteins are captured by antigen-presenting cells (APCs), such as dendritic cells and macrophages. These APCs process the antigen and present the peptide fragments on the cell surface via MHC class II molecules. Once on the surface, the peptide-MHC class II complex is recognized by specific receptors on CD4^+^ T helper cells. This recognition is crucial for the activation of T helper cells, which in turn release cytokines that stimulate other components of the immune system. Among these responses, activated T helper cells promote the differentiation of B cells into plasma cells, leading to the production of specific antibodies against the antigen presented. In addition, T helper cells also activate other immune cells, such as macrophages and cytotoxic T cells, promoting a broader and more coordinated response [73]. Additionally, memory cells are also generated, which are essential for vaccines [69]. The mechanism of action of next-generation vaccines can be seen in Figure 3.

### 4.1. Vaccines Based on Recombinant Proteins

Several protein candidates were described as components of a recombinant antigen vaccine for animal trypanosomiasis. The IFX protein is an invariant glycoprotein recently described in *T. vivax* with great potential to be used as a subunit vaccine. The IFX protein has a flagellar localization and the IFX gene has no paralog and no homology to any other sequence found in trypanosome genome databases. This protein was identified after an extensive study of the genome of the Y468 strain available in TriTrypDB “https://www.tritrypdb.org accessed on 2 September 2024” that included the prediction of GPI-anchored and secreted proteins in this parasite. Sequences from twenty-three proteins were selected to be expressed in eukaryotic cells and evaluated as vaccine candidates. Using an infection model with *T. vivax* expressing luciferase, the authors demonstrated that the IFX protein has the greatest potential as a vaccine candidate, eliciting a long-lasting immune response in BALB/c mice and suggesting the possibility of inducing sterile protection. It was also demonstrated the essential role of antibody-mediated protection that includes complement-mediated lysis [63]. The same group also identified a new family of transmembrane proteins encoded by 124 paralogous genes. These highly immunogenic proteins, named Vivaxin, induce a robust immune response in mice that significantly reduces parasite load, although they do not confer sterile protection. Furthermore, at least one member of the Vivaxin family is expressed on the surface of the bloodstream trypomastigotes of *T. vivax*. Thus, along with VSG, Vivaxin is likely an abundant component of the surface of *T. vivax* and constitutes a potential vaccine candidate [74].

Two potential vaccine candidates recently described for *T. b. brucei*, enolase and ISG75, constitute two examples of antigens that, despite being immunogenic and accessible to the immune system, failed to induce a protective response. Enolase is a highly conserved enzyme that is recognized by host antibodies during infection, while ISG75 is a membrane-anchored, trypanosome invariant surface protein. Although these proteins were highly immunogenic and accessible to the immune system when tested as recombinant antigens in immunization protocols followed by challenge with *T. b. brucei* or *T. evansi*, they failed to induce protection in mice [75].

Paraflagellar rod proteins (PRF) are non-variable proteins that were also described as vaccine candidates for *T. evansi*. Swiss albino mice were immunized with two PFR proteins, TePRF1 and TePRF2, individually and in combination with the two recombinant proteins. The vaccine candidate showed a specific humoral response, including IgG1, IgG2a, and IgG2b, and a cytokine-mediated cellular response. After the lethal challenge with *T. evansi*, previously recovered from a horse, parasitemia was better controlled and survival was extended in immunized mice compared to controls [76]. Similar results were described with recombinant β-tubulin protein from *T. evansi*, when used as an immunizer in mice. The protein induced a specific humoral response with a predominant IgG2a isotope and high levels of IFN-γ, which indicates a Th-1 protective response. After the lethal challenge with *T. evansi*, extended survival and better control of infection were observed in immunized animals compared to the control group [77]. Because actin has a high degree of homology among trypanosomes, this protein was tested as a vaccine candidate that may induce protection against several different trypanosome species. After being immunized with recombinant *T. evansi* actin and challenged with *T. evansi*, *T. equiperdum*, and *T. b. brucei*, protection levels of 63.3%, 56.7%, and 53.3%, respectively, were shown in immunized animals. Although the humoral and cellular responses were not evaluated, the data suggest the importance of the humoral response because the serum collected from immunized rabbits inhibited the growth of the trypanosomes in a dose-dependent manner [78].

### 4.2. The mRNA Vaccines

Due to the challenges in developing vaccines for trypanosomes and the low number of studies showing highly efficient vaccine candidates, new approaches are being considered. The COVID-19 pandemic has brought great advances in vaccine studies, particularly regarding the use of non-replicative viral vectors and RNA vaccines. RNA vaccines have proven to be more effective and faster to produce compared to conventional vaccines because mRNA can be synthesized quickly in the laboratory without needing to express and purify a recombinant antigen in mammalian cells or bacteria. Significant advances have been made in stabilizing mRNA and formulating the lipid nanoparticles surrounding it, facilitating storage and transport. In addition, lipid nanoparticles optimize antigen presentation and T-cell activation [79]. Several authors have recently discussed how RNA vaccines could help the development of a vaccine for *T. cruzi*, which causes human Chagas disease, since the development of conventional vaccines has failed [80]. A vaccine using a homologous and heterologous mRNA/protein protocol with the Tc24 protein, a flagellar protein of *T. cruzi*, was tested. The homologous protocol vaccine was evaluated in C57Bl/6 mice and demonstrated immunogenicity; however, when combined with the heterologous vaccination, it induced significantly higher levels of activated CD4^+^ and CD8^+^ T cells, an increase in cytokine-producing CD4^+^ and CD8^+^ T cells, and a balanced cytokine secretion profile of Th1/Th2/Th17. This suggests that combining protocols may offer a promising vaccine solution targeting trypanosomatids [81].

### 4.3. Vector-Based Vaccines

Vaccines based on recombinant viral vectors also represent a promising new approach for the control of animal trypanosomiasis. In these vaccines, viral vectors are modified to express trypanosome antigens, stimulating a protective immune response in the host. To overcome the problems related to antigenic variation in trypanosomes, one of the main barriers to the development of effective vaccines, the vector-based vaccines may target conserved antigens or crucial virulence factors, such as enzymes involved in immune evasion or tissue invasion. Using viral vectors, trypanosome antigens are delivered in a way that simulates a natural infection, promoting a robust and long-lasting immune response. Like mRNA vaccines, viral vectors can induce high immunogenicity without the use of an adjuvant. This approach also makes it possible to create multivalent vaccines, where different antigens from various Trypanosoma species can be administered in a single dose, extending protection against multiple forms of the disease. Although still in the experimental phase, the use of viral vectors, such as adenoviruses or modified vaccinia viruses, offers several advantages, one of those being the capacity of eliciting both a potent antibody response as well as strong cellular responses [82]. During the COVID-19 pandemic, vaccines using vectors have been licensed: AstraZeneca/Oxford uses a chimpanzee adenovirus (ChAdOx1), modified to carry the Spike protein gene from SARS-CoV-2 [83]; and the vaccine developed by Johnson & Johnson uses a modified human adenovirus (Ad26) to carry the same Spike protein gene, as reviewed by Custers et al. (2021) [84].

### 4.4. CRISPR-Attenuated Vaccines

Few attempts have been described using live attenuated parasites as a vaccine strategy for animal trypanosomiasis. One study showed that calf immunization with *T. evansi* attenuated by γ radiation results in increased titers of serum IgG and IgM and the up-regulation of IFN-γ, TNF, IL-1β, IL-2, and IL17 cytokines. After a lethal challenge with virulent *T. evansi*, the ability of irradiated trypanosomes to confer protective immunity in calves was demonstrated [85].

The use of gene editing is a major breakthrough in the field of vaccines, since it can provide a powerful new genetic tool to generate attenuated trypanosome parasites. As shown with different species of radiation-attenuated parasites, genetically attenuated live parasites are not capable of causing infection because they fail to fully replicate but are still capable of inducing a highly efficient immune response [86,87]. CRISPR allows the precise editing of the pathogen genome in order to eliminate or inactivate one or more genes responsible for virulence. Because it is a highly efficient mechanism of gene deletion, CRISPR-attenuated parasites may be used to target various loci to be deleted in the parasite genome, reducing the risk of reversion to the virulent form, a common concern with traditional attenuated vaccines, making the vaccine safer. Unlike vaccines based on specific subunits or recombinant proteins, live attenuated vaccines can present a wider range of antigens to the immune system, inducing a more robust and long-term immune response, both humoral and cellular. The use of CRISPR protocols is particularly useful to generate live attenuated Trypanosoma vaccines, which have multigene families encoding virulence factors responsive for immune evasion mechanisms [88,89]. CRISPR-attenuated *T. cruzi* has been recently described by our group, in which several copies of trans sialidases (TS) genes, a well-known virulence factor, were disrupted. Although they were able to infect cells in vitro, the attenuated parasites were completely unable to infect mice, even immunodeficient IFN-γ knockout mice. When used to immunize BALB/c mice, the animals were fully protected against a challenge with the virulent *T. cruzi* Y strain. Because they are encoded by a multigene family, knocking out active trans sialidases was only possible with a CRISPR protocol [86]. Additional studies with different Trypanosoma species may be on the way and will soon join the efforts for the development of a veterinary Trypanosoma vaccine. A recent study demonstrated the use of the CRISPR protocol to delete the oligopeptidase B gene in *T. congolense*, which is a virulence factor associated with the parasite capacity of crossing endothelial barriers in microvascular vessels. This is another proof-of-concept study that could not only lead to a deeper understanding of virulence genes but also will facilitate the development of genetically edited, attenuated parasites to be used as vaccines [90].

## 5. Conclusions

The control of trypanosomiasis in farm animals remains a significant challenge for animal health and the economy in several endemic regions. Traditional diagnostic methods, such as microscopy and serological techniques, have limitations, particularly in cases with low parasitemia, underlining the need for more sensitive and specific methods. The introduction of molecular technologies such as PCR and LAMP represents an important advance, but their widespread implementation still faces obstacles, mainly due to high costs and the need for technical training. In parallel, the development of vaccines for trypanosomes that affect livestock faces notable barriers, mainly due to the antigenic variability of the parasites and their capacity for immunosuppression. The limitation regarding suitable laboratory models for various Trypanosoma species is also a major factor that hinders the progress of pre-clinical studies. Despite this, recent advances, such as the identification of invariant proteins and the use of innovative technologies, including mRNA vaccines and gene editing, offer new prospects for the creation of effective vaccines. The success of these efforts depends on overcoming technical and financial challenges, as well as collaboration between public and private initiatives to strengthen research and infrastructure in affected regions. Continued advances in diagnostics and vaccines are therefore fundamental to mitigate the impacts of trypanosomiasis, whereas the adoption of innovative approaches may finally make it possible to effectively control these diseases.

## Figures and Tables

**Figure 1 microorganisms-12-02575-f001:**
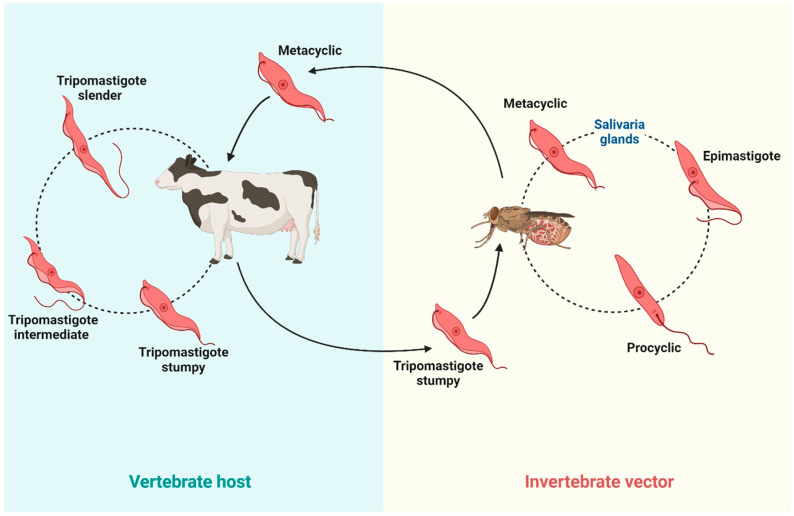
The life cycle of trypanosomes of cattle interest. Vertebrate hosts are bitten by hematophagous flies and infected with the metacyclic form. Inside the host, the trypomastigotes differentiate into three forms: slender, intermediate, and stumpy. When a fly bites an infected animal, it ingests the bloodstream stumpy form of trypomastigotes, which differentiate into proliferative procyclic forms. Once in the digestive tract, procyclic forms multiply by binary fission. In the salivary glands or the proboscis, procyclic forms differentiate into metacyclic forms, starting a new cycle. (Created with BioRender.com).

**Figure 2 microorganisms-12-02575-f002:**
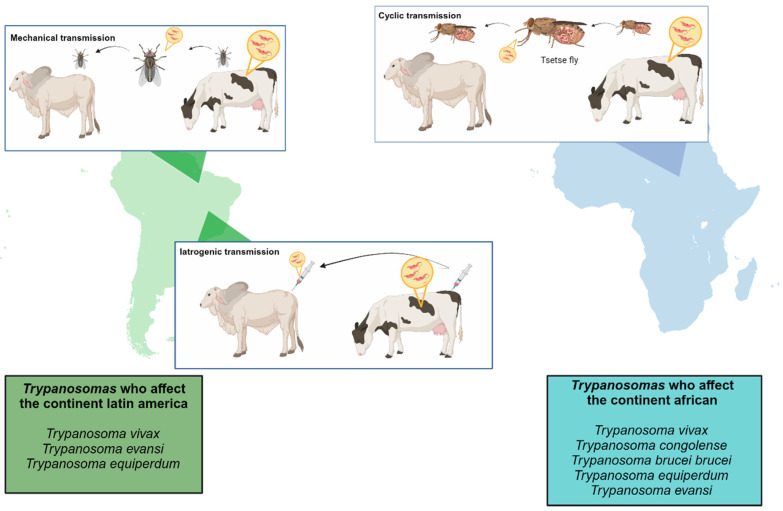
Transmission mechanism between trypanosomes on the African and Latin American continents. Cyclic transmission occurs on the African continent through the tsetse fly, mechanical transmission occurs mainly in Latin America and the Caribbean; and iatrogenic transmission occurs through the use of syringes that have previously been used on infected animals. (Created with BioRender.com).

**Figure 3 microorganisms-12-02575-f003:**
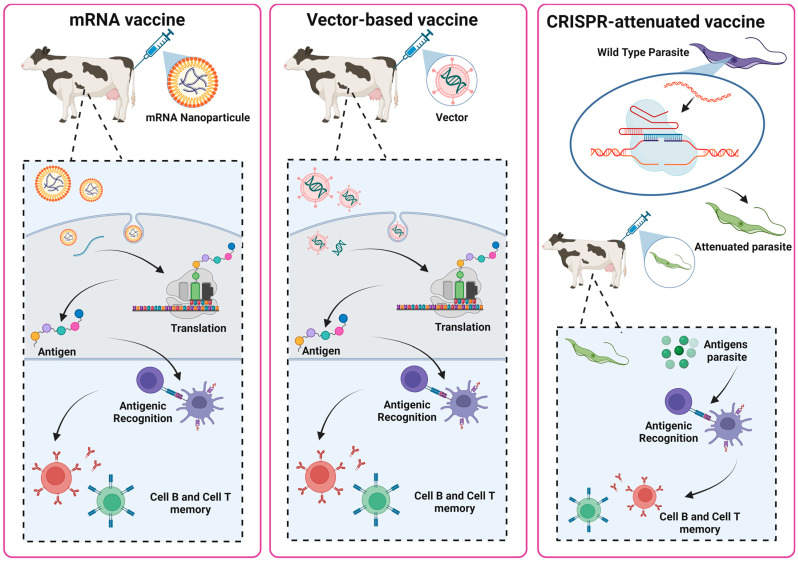
New promising vaccine strategies for animal trypanosomes. RNA vaccines are composed of lipid nanoparticles that protect the mRNA, facilitate mRNA delivery, and activate the host innate immunity. Upon entering the cell, the mRNA is translated and the immune system recognizes the antigen produced, leading to an antibody and T cells’ specific responses to the infectious agent. Vector-based vaccines use a virus or other modified microorganism (vector) to introduce genetic material into the body, stimulating an immune response without causing the disease. The vector carries genes that encode pathogen antigens, prompting the immune system to produce antibodies and specific T cells against the infectious agent. CRISPR-attenuated vaccines use CRISPR-Cas gene editing technology to modify the genome and generate attenuated parasites, making them safe for use in vaccines. The CRISPR technique allows precise modification of genes essential for pathogen virulence, eliminating the parasite’s ability to cause disease without compromising its ability to stimulate a strong immune response. Because they mimic natural infection, these vaccines can generate strong and long-lasting immunity. (Created with BioRender.com).

**Table 1 microorganisms-12-02575-t001:** Animal trypanosomes and their hosts. The main hosts of trypanosomiasis are cattle and horses; however, these trypanosomes can infect other animals, such as sheep, pigs, goats, and camelids.

Trypanosoma	Main host	Other hosts
*Trypanosoma vivax*	Bovine	Sheep Goat Equine Pig
*Trypanosoma congolense*	Bovine	Sheep Goat Equine Pig
*Trypanosoma brucei brucei*	Bovine	Sheep Goat Pig
*Trypanosoma evansi*	Equine	Bovine Camelids Goat Sheep Pig
*Trypanosoma equiperdum*	Equine	-

## Data Availability

No new data were created or analyzed in this study.

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
