# Peer review of "Animal Trypanosomiasis: Challenges and Prospects for New Vaccination Strategies"

_microorganisms, 2024, doi:10.3390/microorganisms12122575_

Round 1

Reviewer 1 Report

Comments and Suggestions for Authors

Dear authors, this research is very comprehensive and complete. However, I have some minor suggestions that might improve your manuscript:

1. Please include some information in the Abstract section about the types of vaccines discussed in this review.

2. Please provide the Materials and Methods chapter, detailing how the selection of data was performed.

3. Please check all the references, as most of them are not formatted correctly.

Author Response

Dear authors, this research is very comprehensive and complete. However, I have some 
minor suggestions that might improve your manuscript:
1. Please include some information in the Abstract section about the types of vaccines 
discussed in this review. 
We are thankful for this suggestion and included in the abstract. (lines 23-24) an indication of the 
vaccine platforms discussed in the review.

2. Please provide the Materials and Methods chapter, detailing how the selection of data was 
performed.
In the previous submission, we did not add a materials and methods section because this is a 
literature review and not a systematic review. However, I believe that a brief paragraph describing 
how the methodology was carried out is welcome. Below is the paragraph that was modified to 
include this information at end of the introduction. (lines 125-133) 
Here, we reviewed the literature with the aim of providing a summary of the major impacts caused 
by these diseases and the current status of the main control methods as well as to discuss the key 
challenges and difficulties faced by researchers involved with the development of vaccines for animal 
trypanosomiasis. We also present new and exciting perspectives for the use of next-generation 
vaccine platforms that could help overcome these barriers. To achieve this, we (i) searched the 
PubMed, ScienceDirect, and Google Scholar databases between September and October 2024 
using the keywords: animal trypanosomiasis, livestock trypanosomiasis, Trypanosoma vivax 
vaccine, Trypanosoma brucei vaccine, Trypanosoma congolense vaccine, Trypanosoma 
equiperdum vaccine and Trypanosoma evansi vaccine; (ii) selected relevant studies published in 
indexed journal articles to discuss their main findings. The selected sources were summarized and 
synthesized in this literature review.

3. Please check all the references, as most of them are not formatted correctly.

All the references have been thoroughly verified

Reviewer 2 Report

Comments and Suggestions for Authors

Journal

Microorganisms

Type of manuscript

Review

Title

Animal Trypanosomiasis: Challenges and Prospects for New Vaccination Strategies

Overview comment:

There is no immediate clarity about the type of review the authors adopt (Scoping Review, Literature Review, Systematic Review). Most reviews clarify and describe in the abstract and introduction the type of review so readers understand the reliability and validity of the review.

The manuscript is presented as “Abstract,” “Introduction,” and “Conclusion.” Normally, there is a section for findings or results but it is not available in this manuscript.

There is no material or methods section. This means there is no description of the type of articles, quality of articles, when the articles were published, and approaches the authors used to identify and check for reliability and validity of articles. Because there is a lack of materials and method section, the thinking process and the actual processes authors used to identify the topics presented and discussed in this manuscript is unclear.

The review goal can be multiple—to support a perspective, to study a research hypothesis and background search of a topic. The goal of a review with defined inclusion and exclusion criteria helps add to the peer-review community with a clearly described and defined methodology so that in the future, another researcher can reproduce the review and continue to add to the richness of the topic. Without this framework, the reproducibility of the review is a challenge.

There are important messages in the manuscript. However, the review methodology and material remain unclear.

Abstract

Clear and align with title. The goal of the manuscript is not immediately clear.

Introduction

The introduction includes the body of the review findings.

It is divided into different subsections. While it is clear by the divisions and sub-headers, it is not clear if the authors are introducing the background of this topic, or describing their review findings, or both. Authors should consider separating introduction from findings.

2. Different trypanosome species that cause animal trypanosomiasis

This appears to be background material. The presentation and writing is clear.

3. Challenges in vaccine development for animal diseases caused by Trypanosomes

The vaccine development description is clear. Again, the robustness of the review methodology is unclear. It will help readers understand the author’s intention if there can be (1) a materials and method section that describes the review approach, inclusion, and exclusion criteria. For review guidelines, please refer to the PRISMA guidelines. (2) If the author can separate the introduction and findings.

Conclusion

The conclusion writing is clear. Some points may err on discussion of the matter, and authors can consider extracting these points for discussion. For example, financial challenges and private and public partnerships are discussions rather than conclusions from findings. Overall, the writing is acceptable as a conclusion for this review.

Author Response

Overview comment:
There is no immediate clarity about the type of review the authors adopt (Scoping Review,
Literature Review, Systematic Review). Most reviews clarify and describe in the abstract and
introduction the type of review so readers understand the reliability and validity of the review.
Following this reviewer’s suggestions, we have added this information in the abstract (line 21) and
introduction (line 135).

The manuscript is presented as “Abstract,” “Introduction,” and “Conclusion.” Normally,
there is a section for findings or results but it is not available in this manuscript.
There is no material or methods section. This means there is no description of the type of
articles, quality of articles, when the articles were published, and approaches the authors
used to identify and check for reliability and validity of articles. Because there is a lack of
materials and method section, the thinking process and the actual processes authors used
to identify the topics presented and discussed in this manuscript is unclear.
The review goal can be multiple—to support a perspective, to study a research hypothesis
and background search of a topic. The goal of a review with defined inclusion and exclusion
criteria helps add to the peer-review community with a clearly described and defined
methodology so that in the future, another researcher can reproduce the review and continue
to add to the richness of the topic. Without this framework, the reproducibility of the review
is a challenge. There are important messages in the manuscript. However, the review
methodology and material remain unclear.
In the previous submission, we did not add a materials and methods section because this is a
literature review and not a systematic review. However, we understood that a brief paragraph
describing how the methodology was carried out would be indeed helpful, please see above the
paragraph that has been added at the end of the introduction, as indicated in our response to the
reviewer #1 (lines 125-133).